# Anthraquinones: Genotoxic until Proven Otherwise? A Study on a Substance-Based Medical Device to Implement Available Data for a Correct Risk Assessment

**DOI:** 10.3390/toxics10030142

**Published:** 2022-03-16

**Authors:** Veronica Cocchi, Sofia Gasperini, Monia Lenzi

**Affiliations:** Department of Pharmacy and Biotechnology, Alma Mater Studiorum University of Bologna, 40127 Bologna, Italy; veronica.cocchi4@unibo.it (V.C.); sofia.gasperini4@unibo.it (S.G.)

**Keywords:** anthraquinones, mutagenicity, in vitro mammalian cell micronucleus test, TK6 cells, ROS, substance-based medical device

## Abstract

A genotoxicological study was carried out on a substance-based medical device (SMD) containing anthraquinones in order to evaluate its potential mutagenic effect. The “In Vitro Mammalian Cell Micronucleus Test” was performed on human TK6 cells by flow cytometry. Cultures were treated with concentrations of SMD tested in the range of 0–2 mg/mL for short treatment time (3 h) both in the absence and presence of an exogenous metabolic activation system, followed by a recovery period in fresh medium (23 h) and for extended treatment time (26 h) without an exogenous metabolic activation system. At the end of both treatment times, cytotoxicity, cytostasis, apoptosis and micronuclei (MNi) frequency were analysed in treated cultures and then compared with those measured in concurrent negative control cultures. The SMD did not induce a statistically significant increase MNi frequency under any of experimental conditions tested. The negative outcome shows that the SMD is non-mutagenic in terms of its ability to induce chromosomal aberrations both in the absence and presence of an exogenous metabolic activation system. The study ended by analyzing intracellular ROS levels to exclude the pro-oxidant ability, typically linked to DNA damage. On the contrary, our results demonstrated the ability the SMD to counteract oxidative stress.

## 1. Introduction

The recent UNI EN ISO 10993-1: 2018 and regulation 2017/745 / EU, which came into force in May 2021, have imposed precise assessments regarding medical devices [1,2]. A complete and exhaustive toxicological analysis must necessarily include the evaluation of the effects on the genetic material in order to exclude the possible genotoxicity, mutagenicity and carcinogenicity of a substance.

The substance-based medical device (SMD) tested in the present study has propulsive and protective activity, resulting in a laxative effect. It was produced as tablets and obtained by a proprietary production process from different botanical species. It contains, among many other bioactives, anthraquinones.

Anthraquinones (9,10-dioxoanthracenes) constitute an important class of natural and synthetic compounds with a wide range of applications. Besides their utilization as colorants, anthraquinone derivatives are employed or studied for a wide array of activities, including laxative, anti-inflammatory, antiarthritic, antifungal, antibacterial, antiviral, antiplatelet, anticancer and neuroprotective effects. However, the presence of quinone moiety in the structure of anthraquinones raises safety concerns, and products that contain them have therefore been under critical reassessment [3].

In particular, on 18 March 2021, the European Commission (EC) published a new regulation (No. 2021/468) on safe consumption of botanical species containing hydroxyanthracene derivatives (HADs), amending Annex III of Regulation (EC) No. 1925/2006 [4]. Pursuant to Article 8 of this regulation, if a non-vitamin, non-mineral substance is associated with a potential risk to consumers, the EC may initiate a procedure to add it, or ingredients containing it, to Annex III, namely those substances whose use in foods is prohibited (Part A), restricted (Part B) or under Union scrutiny (Part C) (Supplementary material 1-SM1). For example, extracts from the leaves of Aloe species containing HADs have been placed on the list of prohibited substances (Annex III, Part A), together with the anthraquinones aloe-emodin, emodin and danthron, due to potentially severe harmful effects on health, including genotoxicity and because the safe daily dose of HADs was unknown. These statements originate from an assessment of the safety of HADs for use in foods carried out by the EFSA and published on 22 November 2017 (EFSA, 2018) [5].

The EFSA Panel on Food Additives and Nutrient Sources Added to Food (ANS) concluded that emodin and aloe-emodin, tested as single substances, have shown reliable evidence of in vitro genotoxicity [6,7,8], whereas rhein and sennosides resulted in non-mutagen in all or the majority of in vitro tests taken into account [8,9]. Furthermore, the Panel deemed aloe-emodin to also be genotoxic in vivo based on the results of a single study obtained by comet assay regarding the pure molecule [10], whereas in several other in vivo experiments performed by Heidemann et al., aloe-emodin did not result in genotoxicity [6]; however, those studies were not considered sufficiently reliable by EFSA [5]. After the EFSA alert, other studies have been published. In particular, recent data proved, by comet assay, that aloe-emodin is not genotoxic in vivo [11].

For sennosides and emodin, only one in vivo study for each substance was considered in the EFSA report, and they demonstrated that they were not genotoxic; however, once again, those outcomes were not considered reliable enough [9,12].

On the other hand, some data suggest an increased risk for colorectal cancer associated with the general use of laxatives, several of which contain HADs, [13], whereas others sustain that the correlation is unproven [14,15]. In light of these ambiguities, the Panel concluded that HADs should be considered genotoxic and carcinogenic unless there are specific data to the contrary, such as for rhein, and that there is a safety concern for extracts containing HADs, although uncertainty persists [5].

Although the SMD tested in our study does not undergo to the regulation of food products, it is advisable to evaluate its safety for use in terms of genotoxicity in consideration of the presence of HADs.

For this reason, in the present study, we evaluated, in vitro, on human lymphoblastoid TK6 cells, the SMD mutagenic potential (in particular in terms of its ability to induce structural and numerical chromosomal aberrations) using the “In Vitro Mammalian Cell Micronucleus Test”, correspondent to OECD guideline n° 487, [16] and analyzed the MNi frequency by a flow cytometric protocol recently developed in our laboratory [17]. Furthermore, in order to exclude a molecular mechanism typically linked to DNA damage, the pro-oxidant ability of the SMD was analyzed.

## 2. Materials and Methods

### 2.1. Reagents

Benzo[a]pyrene (BaP), cyclophosphamide (CP), 2′-7′-dichlorodihydrofluorescin diacetate (DCFH-DA), dimethyl sulfoxide (DMSO), ethylenediaminetetraacetic acid (EDTA), etoposide (ETP), fetal bovine serum (FBS), H_2_O_2_, L-glutamine (L-GLU), mitomycin C (MMC), Nonidet, penicillin-streptomycin solution (PS), phosphate-buffered saline (PBS), potassium chloride, potassium dihydrogen phosphate, Roswell Park Memorial Institute (RPMI) 1640 medium, BPC grade water, sodium chloride, sodium hydrogen phosphate, vinblastine (VINB) (all purchased from Merck, Darmstadt, Germany), Guava Nexin reagent, Guava ViaCount reagent (purchased from Luminex Corporation, Austin, TX, USA), RNase A, SYTOX Green (purchased from Thermo Fisher Scientific, Waltham, MA, USA), and Mutazyme 10% S9 mix (purchased from Moltox, NC, USA).

### 2.2. Test Item

The test item was an SMD produced as tablets by Aboca S.p.A. It is obtained by a proprietary production process from different botanical species reported in Table 1 and expressed as m/m% (i.e., (mass of a single component/total mass) × 100).

#### Anthraquinone Characterization

Anthraquinone content was characterized and quantified by the producer by UHPLC-MS-QToF and is reported in Table 2, expressed as m/m% (i.e., (mass of a single component/total mass) × 100). The compound not quantified is indicated as “nq”, and the compound not detected as “nd”.

At the time of use, tablets were manually crushed using a ceramic mortar to obtain a fine powder, and 200 mg/mL DMSO was used as solvent to prepare the working solution.

The final concentration of DMSO was in the range 0.25–1% in all experimental conditions in order to avoid potential solvent toxicity. 

### 2.3. Cell Culture

OECD guideline n° 487 lists different possible cell lines suitable to analyse the presence of MNi (e.g., CHO, V79, CHL/IU, L5178Y or TK6), among which the most used are TK6 cells, by virtue of their human and non-tumoral origin, ease of maintenance in culture and replicative speed [16,18]. Moreover, this cell line grows in suspension and is therefore an optimal assay system for flow cytometric analyses [19,20,21,22].

TK6 human lymphoblastoid cells were purchased from Merck (Darmstadt, Germany) and were grown at 37 °C under 5% CO_2_ in RPMI-1640 supplemented with 10% FBS, 1% L-GLU and 1% PS. To maintain exponential growth and considering that the time required to complete the cell cycle is about 13 h, the cultures were divided every three days in fresh medium, and the cell density did not exceed the critical value of 9 × 10^5^ cells/mL.

### 2.4. Test Conditions

As recommended by OECD guideline n°487, TK6 cells were treated in the absence and presence of an exogenous metabolic activation system in order to highlight the effects not only of the parental compounds but also of any metabolites produced in vitro. The exogenous metabolic activation system employed is a cofactor-supplemented post-mitochondrial fraction (S9 mix) prepared from the livers of rats treated with enzyme-inducing agent Aroclor 1254. The final concentration of S9 in the culture medium was 1%. This guideline also recommends limiting cell exposure to S9 mix to a short treatment time [16]. To ensure a thorough evaluation of the mutagenic potential, it suggests conducting the experiment at a short treatment time (3 h) both in the absence and presence of S9 mix, followed by recovery in fresh medium for a total period equivalent to about 1.5–2.0 normal cell-cycle lengths after the beginning of treatment (i.e., a total of 26 h for TK6 cells) and at an extended treatment time without metabolic activation equivalent to about 1.5–2.0 normal cell-cycle lengths after the beginning of treatment (26 h for TK6 cells) [16].

#### 2.4.1. Selection of Concentrations

OECD guideline n° 487 establishes a cytotoxicity and cytostasis threshold equal to 55 ± 5% and a consequent viability and proliferation threshold equal to 45 ± 5% to select the suitable concentrations to be tested for MNi frequency evaluation [16]. For this purpose, TK6 cells were treated with different SMD concentrations and for different times, as specified below.

#### 2.4.2. Measurement of Cytotoxicity

Aliquots of 2.5 × 10^5^ of TK6 cells were treated with the SMD at 0, 0.03125, 0.0625, 0.125, 0.25, 0.5, 1.0 and 2.0 mg/mL for 3 h with or without S9 mix followed by 23 h of recovery period and for 26 h without S9 mix. At the end of the treatment time, cytotoxicity was evaluated by Guava ViaCount assay and expressed as viability percentage. In particular, cells were stained with Guava ViaCount reagent (containing propidium iodide, PI) as reported in the relative datasheet, and 1000 cells were analyzed by Guava ViaCount software. The viability percentage recorded in the treated cultures was normalized to that recorded in the concurrent negative control cultures, considered equal to 100%.

#### 2.4.3. Measurement of Cytostasis

Aliquots of 2.5 × 10^5^ of TK6 cells were treated with the SMD at 0, 0.03125, 0.0625, 0.125, 0.25, 0.5, 1.0 and 2.0 mg/mL for 3 h with or without S9 mix followed by 23 h of recovery period and for 26 h without S9 mix. At the end of the treatment time, cytostasis was evaluated by Guava ViaCount assay and expressed as population doubling (PD), calculated as follows:PD=[log(post−treatment cell number÷initial cell number)]÷2

Subsequently, the PD obtained in the concurrent negative control cultures was compared to that measured in the treated cultures, obtaining the relative population doubling (RPD) value to measure the cell proliferation and to check that a substantial proportion of the cells had undergone division at the end of treatment time. RPD was calculated with the following formula:$$\mathrm{RPD}=\frac{\mathrm{PD}\;\mathrm{in}\;\mathrm{treated}\;\mathrm{cultures}}{\mathrm{PD}\;\mathrm{in}\;\mathrm{control}\;\mathrm{cutures}}\times 100$$

#### 2.4.4. Measurement of Apoptosis

To better select the suitable concentrations to be tested for MNi frequency evaluation, we analysed apoptosis as a cell death mechanism alternative to necrosis. Such concentrations induce apoptosis up to a doubling of that recorded in the concurrent negative cultures [16,17].

Aliquots of 2.5 × 10^5^ of TK6 cells were treated with SMD concentrations selected on the basis of the results of cytotoxicity and cytostasis analyses, i.e., 0, 0.03125, 0.0625, 0.125, and 0.25 for 3 h with or without S9 mix followed by 23 h of recovery period and for 26 h without S9 mix. A concentration of 5 µg/mL ETP was used as positive control. At the end of the treatment time, the percentage of apoptotic cells was evaluated by Guava Nexin assay. Cells were stained with Guava Nexin reagent (containing 7-aminoactinomycin, 7-AAD and Annexin-V-PE) as reported in the relative datasheet, and 2000 cells were analyzed by Guava Nexin software. The apoptotic cell percentage recorded in the treated cultures was normalized to that recorded in the concurrent negative control cultures, considered equal to 1, and expressed as apoptotic fold increase.

#### 2.4.5. Measurement of MNi Frequency

Aliquots of 2.5 × 10^5^ of TK6 cells were treated with SMD concentrations selected on the basis of the results of cytotoxicity, cytostasis and apoptosis analysis, i.e., 0, 0.03125, 0.0625, 0.125, and 0.25 for 3 h with or without S9 mix followed by 23 h of recovery period and 0, 0.03125, 0.0625, 0.125 for 26 h without S9 mix. Clastogen MMC and aneugen VINB were used as positive controls active without metabolic activation, whereas the clastogens CP and BaP were used as positive controls requiring metabolic activation [16].

At the end of the treatment time, the MNi frequency was evaluated by a flow cytometric protocol developed in our laboratory and published in [17].

Briefly, cells were collected, lysed and stained with SYTOX Green. The discrimination between nuclei and MNi was performed on the basis of the different size analyzed by forward scatter (FSC) and the different intensity of green fluorescence. The MNi frequency, calculated as the number of MNi per 10,000 nuclei deriving from viable and proliferating cells and recorded in treated cultures at all concentrations tested, was normalized to that recorded in the concurrent negative control cultures, considered equal to 1, and expressed as MNi frequency fold increase.

#### 2.4.6. Measurement of Intracellular ROS Levels

Aliquots of 2.5 × 10^5^ of TK6 cells were treated with the SMD at 0, 0.03125, 0.0625, 0.125, 0.25, 0.5, 1.0 and 2.0 mg/mL both in the absence and presence of H_2_O_2_ for 1 h and 6 h because ROS generation is known to be an early event [23].

At the end of the treatment time, after checking the viability by the Guava ViaCount assay, intracellular ROS levels were evaluated by the fluorescent probe DCFH-DA. Cells were incubated with DCFH-DA 10 μM in PBS 1× for 20 min and then exposed to H_2_O_2_ 100 μM for 20 min. Subsequently, 5000 events derived from viable cells were analyzed by Guava InCyte software.

The fluorescence intensity recorded in the treated cultures was normalized to that recorded in the concurrent negative control cultures, considered equal to 1, and expressed as ROS fold increase.

### 2.5. Flow Cytometry

All FCM analyses reported above were performed using a Guava EasyCyte 5HT flow cytometer equipped with a class IIIb laser operating at 488 nm (Luminex Corporation, Austin, TX, USA).

### 2.6. Statistical Analysis

All assays were performed in triplicate and repeated in three independent experiments.

All results are expressed as mean ± SEM of three independent experiments. Statistical significance was analyzed by paired analysis of variance (repeated ANOVA), followed by Dunnett or Bonferroni post-test using Prism Software 4.

## 3. Results

### 3.1. Measurment of Cytotoxicity

Cytotoxicity has to comply with the threshold equal to 55 ± 5% established by OECD. Consequently, cellular viability must be greater than 45 ± 5% [16].

In the bar charts reported in Figure 1, it can be seen how the viability in SMD-treated cultures complied with the OECD guideline threshold (represented by the red line) up to 0.5 mg/mL concentration at all the experimental conditions. Moreover, in Figure 1, we inserted two FCM dot plots corresponding to a representative negative control culture (left) and a representative culture treated SMD 0.5 mg/mL (right) at 26 h −S9 mix condition to show how the viable and dead cells were visualized and discriminated on the basis of low and high fluorescence intensity, respectively.

### 3.2. Measurement of Cytostasis

Similarly to cytotoxicity, cytostasis has to comply with the threshold equal to 55 ± 5% established by OECD. Consequently, cellular proliferation (expressed as RPD) must be greater than 45 ± 5% [16].

Based on the obtained results, RPD in SMD-treated cultures complied with the OECD guideline threshold up to 0.25 mg/mL after the short treatment time both in the absence and presence of S9 mix and up to 0.125 mg/mL after 26 h treatment, as highlighted by the red line (Table 3).

### 3.3. Measurement of Apoptosis

Based on cytotoxicity and cytostasis results, concentrations up to 0.25 mg/mL for the short treatments and up to 0.125 mg/mL for the extended treatment were tested for apoptosis analysis.

The results obtained show that up to 0.125 mg/mL, the apoptosis fold increase measured in the SMD-treated cultures was comparable to that measured in the concurrent negative control culture at all experimental conditions. More specifically, at 0.125 mg/mL, 1.11 ± 0.07 and 1.18 ± 0.15 were measured after short and extended treatment time, respectively, without S9 mix and 1.35 ± 0.22 after short treatment time in the presence of S9 mix.

At 0.25 mg/mL, after short treatment time without S9 mix, a slightly higher increase was detected, but it did not reach a doubling compared to the concurrent negative control (1.68 ± 0.35 vs. 0 mg/mL).

Moreover, in Figure 2, we inserted two FCM dot plots corresponding to a representative negative control culture (left) and a representative culture treated with ETP 5 µg/mL (right) at 26 h −S9 mix condition in order to show how the viable, apoptotic and necrotic cells were visualized and discriminated on the basis of the 7-AAD/Annexin V-PE fluorescence intensity.

### 3.4. Measurement of MNi Frequency

Based on the results obtained from cytotoxicity, cytostasis and apoptosis analyses, the concentrations to be tested in the “In Vitro Mammalian Cell Micronucleus Test” were selected: 0.03125, 0.0625, 0.125 and 0.25 mg/mL for the short treatment time in the absence and presence of S9 mix and 0.03125, 0.0625 and 0.125 mg/mL for the extended treatment time.

The FCM dot plots reported in Figure 3 allow to visualize how the events in the MNi gate are comparable in treated cultures and in the concurrent negative control cultures at all experimental conditions tested. Even more evidently, the bar charts demonstrate that the SMD did not induce any statistically significant increase of MNi frequency (Figure 3).

### 3.5. Measurement of Intracellular ROS Levels

This test was initially performed to exclude the SMD pro-oxidant ability, typically linked to DNA damage. For this purpose, all SMD concentrations in the range of 0–2 mg/mL were tested at two different treatment times, 1 h and 6 h, as ROS generation is known to be an extremely early event.

At the end of the treatment time, cellular viability was checked (Table 4) and ROS intracellular levels were measured.

The results showed an antioxidant activity. Indeed, ROS intracellular levels were significantly decreased in the cultures treated with the SMD at concentrations of 0.25 mg/mL or higher compared to the concurrent negative control cultures at both treatment times (Figure 4, left bar charts). Consequently, to explore the possible effect of the SMD in counteracting oxidative stress, cells were pretreated with the SMD and then exposed to H_2_O_2_. As illustrated in Figure 4 (right bar charts and FCM histograms), the SMD treatment counteracted intracellular ROS production both after 1 h and 6 h treatment times. In particular, a statistically significant decrease in ROS levels was measured in the cultures treated with 0.25 mg/mL, 0.5 mg/mL, 1 mg/mL and 2 mg/mL in the presence of H_2_O_2_ at both treatment times compared to the concurrent positive control culture (Figure 4, right bar charts).

## 4. Discussion

Anthraquinones are compounds of natural origin present in numerous botanical species that are widely used in folk medicines, foods, cosmetics and pharmaceuticals [3]. However, the EFSA ANS Panel established that some HADs “should be considered as genotoxic and carcinogenic unless there are specific data proving the contrary” [5].

In consideration of the fact that the SMD tested is a natural mixture containing anthraquinones, the aim of the present study was to verify its potential genotoxicity.

The European reference standard for the biological evaluation of medical devices is UNI EN ISO 10993-3, and in particular, the genotoxicity tests recommended by most regulatory agencies and international authorities, such as the International Conference on Harmonization (ICH), are listed in the supplement part 33, “Guidance on tests to evaluate genotoxicity” [24,25]. Among these, we selected the “In Vitro Mammalian Cell Micronucleus Test” for two main reasons. First of all, it is a scientifically valid, versatile and effective test that highlights both structural chromosomal aberrations induced by clastogenic agents and numerical chromosomal aberrations induced by aneuploidogenic agents. Furthermore, in 2016, the in vitro version of the test was validated by the OECD (guideline n° 487) [16]. In 2018, we published an innovative flow cytometric protocol to score MNi to overcome some critical issues affecting the classic method of analysis by optical microscopy, such as the subjectivity of interpretation, long samples preparation and analysis times and the number of events examined, which is extremely low for the purpose of a robust statistical analysis. Since then, our protocol was successfully used to prove the mutagenicity of different novel psychoactive substances [20,21,22] and the lack of mutagenic capacity of Wasabia japonica [19].

As recommended by the OECD, we proceeded by treating cells for a short treatment time in the absence and presence of extrinsic metabolic activation (i.e., S9 mix) specifically in order to control the possible mutagenic capacity not only of the parental compounds but also of any metabolite produced in vitro. Furthermore, to ensure a thorough evaluation and to conclude a negative outcome, guideline n°487 also suggests conducting the experiment at an extended treatment time without metabolic activation.

To assess the genotoxicity the first fundamental step is to define the concentrations of the product to be tested, which must not induce marked changes in pH and osmolality of the cell medium and must lead to absent or poor cell death and no or limited inhibition of cell proliferation [16]. For this reason, OECD guideline n°487 established a cytotoxicity and cytostasis threshold equal to 55 ± 5% and consequently recommends proceeding only if the treated population shows a cell viability and a cell proliferation of at least 45 ± 5% when compared to concurrent negative control cultures [16].

For this purpose, we performed a Guava ViaCount assay to evaluate, by flow cytometry, cell viability of untreated and treated cultures with SMD scalar concentrations from 0 to 2 mg/mL, corresponding to the maximum concentration allowed by the OECD guideline. The results obtained show that the SMD complied with the threshold of up to 0.5 mg/mL concentration at all experimental conditions.

The same assay also allowed us to carry out a robust measurement of the number of cells at the time of cellular seeding (time zero) and at the end of the treatment time, which is fundamental for checking, with great accuracy, the correct cell proliferation. In this case, the threshold was respected up to 0.25 mg/mL concentration at the short treatment time with or without S9 mix and up to 0.125 mg/mL concentration at the extended treatment time. The limited cytotoxicity and cytostasis could be considered positive results, but this is not entirely true from a genotoxicological point of view. Indeed, a substance capable of damaging DNA but allowing the cell population to survive and replicate also makes it able to transmit any genetic damage to its offspring.

The Guava ViaCount assay enables an effective discrimination between viable and necrotic cells but does not highlight apoptotic cells. In fact, the distinction is based solely on the different membrane integrity and consequent permeability to the dye. For this reason, apoptotic cells, characterized by a still-intact membrane, could be “confused” by the instrument for living cells. Therefore, we considered it necessary to proceed with a more specific test to highlight this alternative death mechanism, which is particularly important for genotoxicity. In fact, the cell population exposed to a genotoxic agent could be stimulated to undergo apoptosis following unrepaired genetic damage. On the contrary, resistance to apoptosis can result in the inability of cells to counteract, through this mechanism of selective death, the transmission of genetic damage to the daughter cells. For this reason, we performed a Guava Nexin assay to evaluate, by flow cytometry, the apoptosis levels of untreated and treated cultures with SMD concentrations selected on the basis of the results obtained in the previous assays of cytotoxicity and cytostasis (i.e., concentrations up to 0.25 mg/mL for the short treatments and up to 0.125 mg/mL for the extended treatment). The results obtained with double staining 7-AAD/Annexin V-PE highlighted that the SMD up to 0.125 mg/mL determined an apoptosis fold increase comparable to that measured in the concurrent negative control culture at all experimental conditions. At 0.25 mg/mL after a short treatment time without S9 mix, a slightly higher increase was detected, but it did not reach a doubling compared to the concurrent negative control.

OECD guideline n °487 lists different MNi scoring procedures, among which we selected flow cytometry because it offers numerous advantages compared to optical microscopy, such as greater objectivity and statistical robustness of the results, with a significant reduction in analysis times. The flow cytometric protocol developed in our laboratory [17] permitted us to demonstrate the inability of the SMD to statistically significantly increase the MNi frequency at all the experimental conditions.

The obtained negative outcome proves the inability of the SMD to induce structural and numerical chromosomal aberrations both in the presence and absence of an exogenous metabolic activation system. These results are not in contrast with some data reported in the literature, which state that aloe-emodin and emodin, also present in the tested SMD, are genotoxic [6,7,8,10]. Indeed, in the present study, a complex mixture of substances was tested, as opposed to single molecules; this could suggest a possible involvement of the matrix effect. In this regard, the “Guidance on safety assessment of botanicals and botanicals preparations intended for use as ingredients in food supplements” published by EFSA states, “It is plausible that the kinetics as well as the expression of the inherent toxicity of a naturally occurring substance could be modified by the matrix in which it is present. Depending on the mechanism of action, this could result in the toxicity being unchanged, reduced or even increased. Where a matrix effect is advocated to support the safety of specific levels of substances (e.g., that data from a pure substance may overestimate effects of the substance in the botanical matrix), testing and/or other data should be provided to demonstrate the occurrence of the matrix effect of the preparation and its magnitude. A matrix effect should be judged on a case-by-case basis” [26].

A further consideration must be made in relation to the fact that the “In Vitro Micronucleus Test on Mammalian Cells” allows for evaluation of whether the tested product is capable of inducing chromosomal aberration and aneugenicity, although “chemicals can induce genetic damage by different mechanisms, so a battery of test sensitive to a different type of genetic damage are thought to provide the best assurance for detecting genotoxic hazard”. In particular, for medical devices, UNI EN ISO 10993-33 identified the following strategy: a test to identify chromosomal aberration and aneugenicity, such as the MN test, and a bacterial reverse mutation test to identify point mutation [24]. Therefore, the negative outcome obtained in our study represents only the first but important step to hypothesize that SMD is not mutagenic. The next natural step is to proceed with further studies, performing a bacterial reverse mutation test in order to completely exclude its mutagenic capacity.

The research conducted in the present work ended by analyzing intracellular ROS levels in order to exclude the SMD pro-oxidant ability, which represents an event typically linked to DNA damage. In fact, it has long been known how ROS, such as ^1^O_2_, O_2_^•−^, H_2_0_2_ and ^•^OH, are able to attack DNA and how oxidative stress is potentially involved in genetic damage [21,27,28]. Our results showed an opposite effect. Not only was the SMD not responsible for increased ROS production; on the contrary, the DCFH-DA assay demonstrated a statistically significant decrease in ROS levels in treated cultures compared to the concurrent negative control cultures. These results suggested the next natural step, which is to stress the cells with the well-known pro-oxidant H_2_O_2_ after having pre-treated them with the test item to check whether it has real antioxidant capacity.

The obtained outcomes strongly confirm this hypothesis; in fact, a dose- and time-related intracellular ROS decrease was observed, demonstrating the SMD’s ability to counteract oxidative stress. These observations are in agreement with what is reported in the literature for some natural anthraquinones from plants [29,30]. In particular, aloe-emodin and emodin are primarily known for their laxative power, although they also exhibit an important antioxidant effect. In particular, aloe-emodin exerts a reducing activity and scavenges hydroxyl radicals; emodin is also a good scavenger of free-radical species [31,32,33].

Flavonoids are also present in the botanical species making up the complex mixture at the base of the SMD, beneficial effects of which, also due to their antioxidant activity, are well known [31,34,35,36]. More specifically, these molecules can act through different antioxidant mechanisms, including suppression of ROS formation, either by inhibition of enzymes or by chelating trace elements involved in free-radical generation, ROS scavenging and upregulation or protection of antioxidant defenses [34,37,38,39].

Finally, it is important to consider that the present study was conducted in vitro. The results obtained may need to be confirmed in vivo, but opinions about it are controversial [24].

## 5. Conclusions

Research conducted in the present study provides a valid basis to support the SMD’s inability to induce structural and numerical chromosomal aberrations, as well as its antioxidant power.

## Figures and Tables

**Figure 1 toxics-10-00142-f001:**
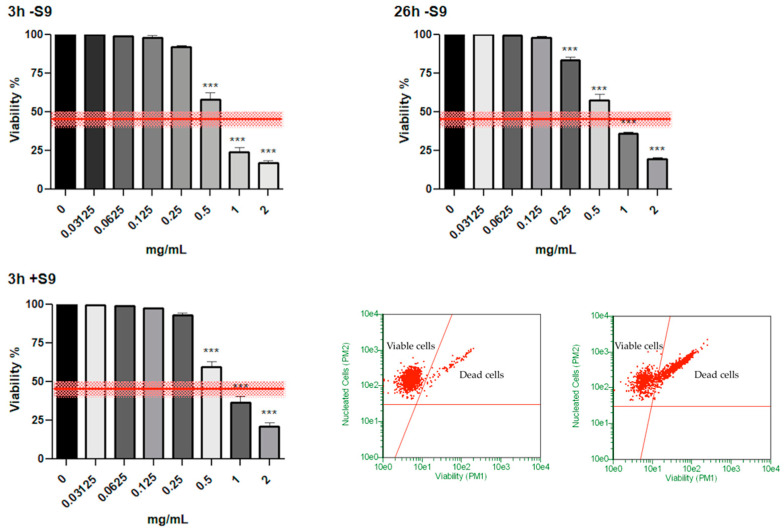
Cell viability on TK6 cells after 3 h −S9 mix, 26 h −S9 mix and 3 h +S9 mix treatment with the SMD at the indicated concentrations compared to the concurrent negative control (0 mg/mL). Each bar represents the mean ± SEM of at least three independent experiments. Data were analysed using repeated ANOVA, followed by Dunnet or Bonferroni post-tests. *** *p* < 0.001 vs. (0 mg/mL). The red lines in the bar charts represent the OECD threshold for viability (45 ± 5%). Lower right: representative FCM dot plots of viable and dead cells after 26 h S9 mix in culture treated with SMD 0.5 mg/mL (right dot plot) and the concurrent negative control culture (left dot plot).

**Figure 2 toxics-10-00142-f002:**
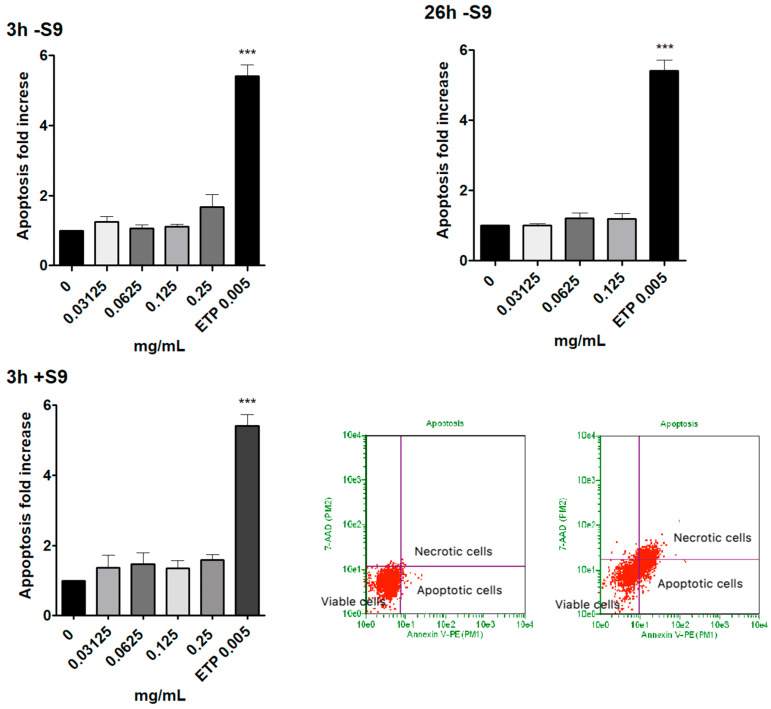
Apoptosis fold increase on TK6 cells after 3 h −S9 mix, 26 h −S9 mix and 3 h +S9 mix treatment with the SMD or ETP at the indicated concentrations compared to the concurrent negative control (0 mg/mL). Each bar represents the mean ± SEM of at least three independent experiments. Data were analysed using repeated ANOVA, followed by Dunnet post-test. *** *p* < 0.001 vs. (0 mg/mL). Lower right: representative FCM dot plots of viable, apoptotic and necrotic cells after 26 h −S9 mix in culture treated with ETP 5 µg/mL (right dot plot) and in the concurrent negative control culture (left dot plot).

**Figure 3 toxics-10-00142-f003:**
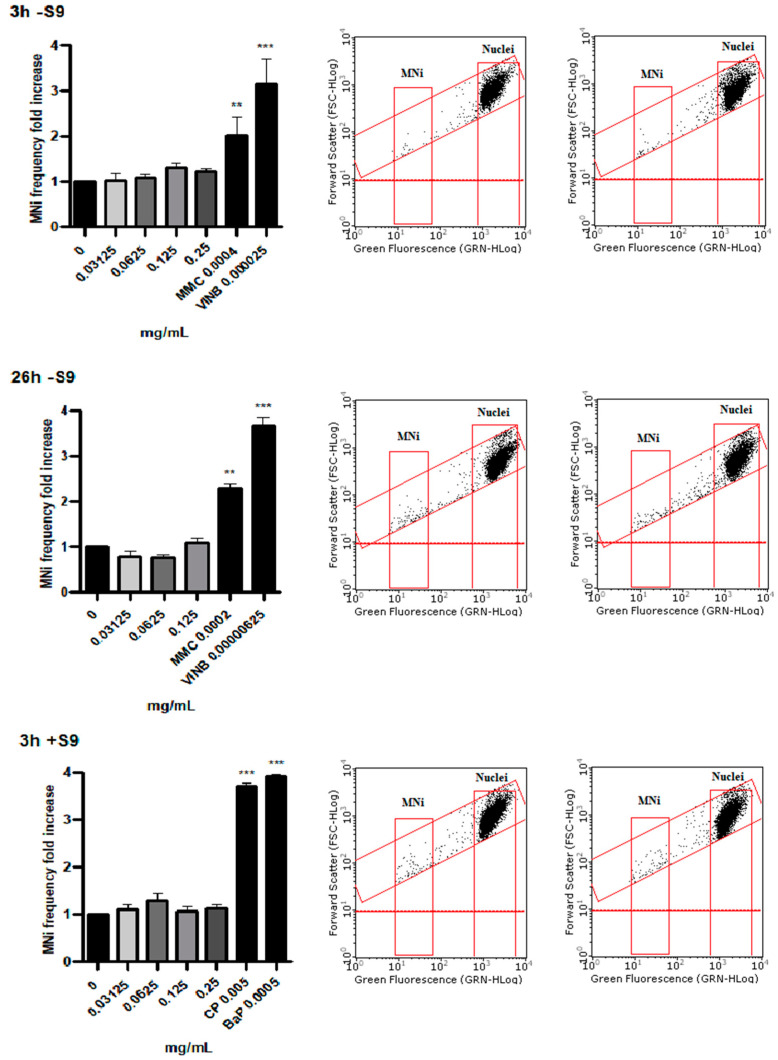
MNi frequency fold increase on TK6 cells after 3 h −S9 mix, 3 h +S9 mix and 26 h −S9 mix treatment with the SMD or positive controls (MMC, VINB, CP or BaP) at the indicated concentrations compared to the concurrent negative control (0 mg/mL). Each bar represents the mean ± SEM of at least three independent experiments. Data were analysed using repeated ANOVA, followed by Dunnet post-test. ** *p* < 0.01 vs. (0 mg/mL); *** *p* < 0.001 vs. (0 mg/mL). Representative FCM dot plots of nuclei and MNi in cultures treated with SMD 0.125 mg/mL (right dot plot) and in the concurrent negative control culture (left dot plot).

**Figure 4 toxics-10-00142-f004:**
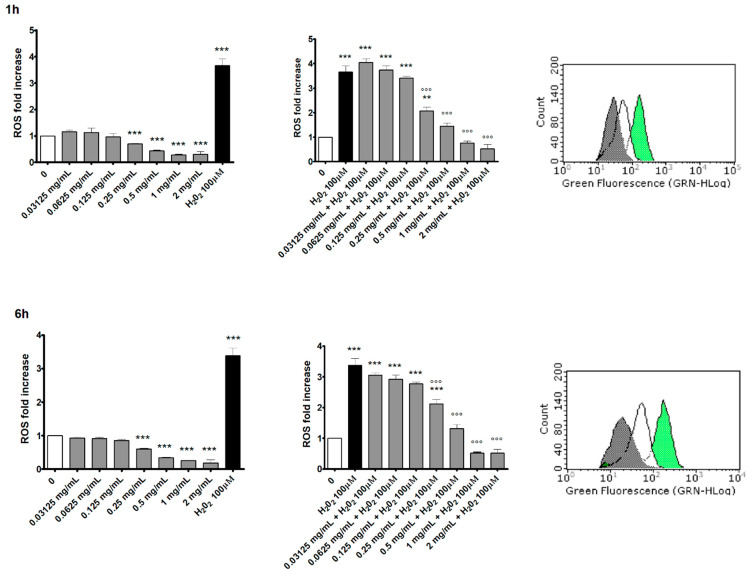
ROS fold increase on TK6 cells after 1 h and 6 h treatment with the SMD and/or positive control (H_2_O_2_) at the indicated concentrations compared to the concurrent negative control (0 mg/mL). Each bar represents the mean ± SEM of at least three independent experiments. Data were analysed using repeated ANOVA, followed by Dunnet post-test. ** *p* < 0.01 vs. (0 mg/mL); *** *p* < 0.001 vs. (0 mg/mL). °°° *p* < 0.001 vs. (H_2_O_2_ 100 μM). Representative FCM histograms of SMD 1 mg/mL + H_2_O_2_ (grey), concurrent negative control (white) and H_2_O_2_ (green) cultures.

**Table 1 toxics-10-00142-t001:** Botanical composition of SMD tablets expressed as m/m% (i.e., (mass of a single component/total mass) × 100).

Botanical Species	Part of the Plant Used	m/m%
*Cassia angustifolia*	Leaves, dry extract	29.8%
*Cichorium intybus*	Roots, powder	27.151%
*Cassia angustifolia*	Leaves, powder	23.801%
*Carum carvi*	Seeds, powder	5.762%
*Taraxacum officinale*	Roots, dry extract	5%
*Foeniculum vulgare*	Seeds, powder	4.84%
*Foeniculum vulgare*	Seeds, dry extract	2.1%
*Cuminum cyminum*	Seeds, powder	1.546%

**Table 2 toxics-10-00142-t002:** Anthraquinone content of SMD tablets expressed as m/m% (i.e., (mass of a single component/total mass) × 100).

Compound	m/m%
Aloe-emodin	0.017855
Emodin	0.003442
Emodin-8-glucoside	nq
Rhein	0.049077
Rhein-8-glucoside	0.115634
Sennidine B	nd
Sennoside A	0.627051
Sennoside A1	0.068689
Sennoside B	0.386504
Sennoside C	0.219383
Sennoside D	0.056659
Anthraquinones, total	1.5443

**Table 3 toxics-10-00142-t003:** RPD on TK6 cells after 3 h −S9 mix, 26 h −S9 mix and 3 h +S9 mix treatment with the SMD at the indicated concentrations compared to the concurrent negative control (0 mg/mL). Each value represents the mean ± SEM of at least three independent experiments. Data were analysed using repeated ANOVA, followed by Dunnet post-tests. ** *p* < 0.01 vs. [0 mg/mL]; *** *p* < 0.001 vs. (0 mg/mL). The red lines highlight up to which concentration the OECD threshold for cellular replication is respected.

RPD 3 h −S9		RPD 26 h −S9		RPD 3 h +S9	
0 mg/mL	100%	0 mg/mL	100%	0 mg/mL	100%
0.03125 mg/mL	90.03 ± 5.83%	0.03125 mg/mL	97.86 ± 2.14%	0.03125 mg/mL	93.90 ± 4.69%
0.0625 mg/mL	90.66 ± 6.19%	0.0625 mg/mL	94.64 ± 3.22%	0.0625 mg/mL	78.75 ± 6.97%
0.125 mg/mL	76.69 ± 10.19% **	0.125 mg/mL	77.52 ± 2.55% ***	0.125 mg/mL	80.57 ± 4.73%
0.25 mg/mL	57.74 ± 6.19% ***	0.25 mg/mL	31.13 ± 8.44% ***	0.25 mg/mL	49.55 ± 9.33% **
0.5 mg/mL	0 ± % ***	0.5 mg/mL	0 ± % ***	0.5 mg/mL	0 ± % ***
1 mg/mL	0 ± % ***	1 mg/mL	0 ± % ***	1 mg/mL	0 ± % ***
2 mg/mL	0 ± % ***	2 mg/mL	0 ± % ***	2 mg/mL	0 ± % ***

**Table 4 toxics-10-00142-t004:** Cell viability on TK6 cells after 1 h or 6 h treatment with the SMD at the indicated concentrations compared to the concurrent negative control (0 mg/mL). Each value represents the mean ± SEM of three independent experiments.

Viability 1 h		Viability 6 h	
0 mg/mL	100%	0 mg/mL	100%
0.03125 mg/mL	97.39± 2.03%	0.03125 mg/mL	100 ± 1.00%
0.0625 mg/mL	98.14 ± 1.9%	0.0625 mg/mL	99.45 ± 0.55%
0.125 mg/mL	99.82 ± 0.19%	0.125 mg/mL	100 ± 1.31%
0.25 mg/mL	98.80 ± 0.03%	0.25 mg/mL	99.25 ± 0.25%
0.5 mg/mL	97.08 ±0.76%	0.5 mg/mL	99.28 ± 0.73%
1 mg/mL	99.51 ± 0.15%	1 mg/mL	98.90 ± 0.60%
2 mg/mL	99.69 ± 0.32%	2 mg/mL	98.40 ± 0.60%

## Data Availability

Not applicable.

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
