# Peer review of "Anthraquinones: Genotoxic until Proven Otherwise? A Study on a Substance-Based Medical Device to Implement Available Data for a Correct Risk Assessment"

_toxics, 2022, doi:10.3390/toxics10030142_

Round 1
Reviewer 1 Report
brief summary of the article and its contribution
General remarks:
Overall, the paper is well-organized and contains all of the required components of a scientific research article. However, would benefit from rationale and reasoning. Most importantly, the methods and materials section should not just be a list of the components utilized, but should also explain why certain decisions were taken. For example, for each of the in vitro tests, a description of the examined concentration range is missing. Another example is the absence of a discussion of the selection of in vitro tests to assess mutagenicity in comparison to other tests employed.
The authors claim to answer the question on the genotoxicity of their MDBS. Their outspoken claims should be viewed in the light of the versatility of the genotoxicity phenomenon. By performing and executing only one in vitro test no conclusive answer can be given on this matter. Considering the Notes of Guidance for testing a cosmetic product, investigating the mutagenicity of a certain chemical is only half of the puzzle. A subsequent Bacterial Reverse mutagenesis test should be performed and also be negative to further conclude the negative genotoxicity of the chemical. Only when converse outcomes are perceived a third test, in vitro, can with sufficient accuracy clear out the matter. As a result, while we appreciate the study group's well-executed experimental effort, we are dissatisfied with the impending conclusions. Some extra research is required to back up the assertions stated in the paper under review, and this will only strengthen or extend the work.
Major in-depth remarks:
Certain critical sections lack reasoning or motivation, preventing the reader from fully understanding the researcher's decisions and making the article more understandable. For example, the absence of cell viability assessments after 26 hours using S9 mix. Concentrations tested for each in vitro assay (cell viability, MNVit, Ros). In general, an overview or critical discussion of the MNVit test's selection as an in vitro technique to study mutagenicity (perhaps other studies have been performed on MDBS with this test).
In addition to the above comments on the motivation for particular decisions, the choice of ROS test doses is disputed. MDBS concentrations are evaluated up to 1 mg/ml; nevertheless, the preliminary results on cell viability (+ figure 1) show that cell viability reduces at doses greater than 0.5. In light of this, we fear on the interpretation of these results. Perhaps there is a reasonable explanation for this, but it is currently lacking.
Minor in-depth remarks
Abbreviations should be defined throughout the text, even in the subtext of a figure or table. The majority of the abbreviations are explained in the Materials and Methods section, however this is stated at the bottom of the manuscript. One approach is to write the term lengthy the first time and only abbreviate it later. Also, numerous abbreviations, such as nq, nd, and percent p/p, are not defined throughout the article. I recommend that each abbreviation referenced in the text should be completely revised.
Use of latin words e.g. in vitro should be in italic.
Technically, the term "medical device based on substances" is correct; but, in written text, it is too long and disrupts the flow of the sentence. In addition, this term is not utilized in a legal setting. As a result, I propose that the term "substance-based medical device" could be used throughout the text. This terminology is also used by the European commission. This should also be adapted in the title.
According to OECD criteria, the cellular viability percentage is 55 percent. The figure 1 Cell viability, on the other hand, shows a red line at the 50% threshold. This should be changed, and the red line should be described in the text beneath the figure.
Figures are accompanied by flow cytometric dot plots, but these are not mentioned in the results or discussion sections. This should either be implemented or used as supplemental material.
Figure 2, the x-axis of the 3h + S9 mix appears introverted. As a result, it can be misinterpreted and cause confusion.
Figure 3 indicates the statistical level with a "*"- sign. However, only "**" and "***" are visible in figure 3. As a result, an explanation of the "*"-symbol is unnecessary here.

Reviewer 2 Report
This manuscript evaluated the cytotoxic and genotoxic effects of a mixture containing anthraquinones using TK6 cells. The genotoxicity was evaluated using a standard genotoxicity assay, the micronucleus (MN) assay following the OECD guideline No. 487, which is good. The authors conclude that the test article is non-mutagenic in terms of its ability to induce chromosomal aberrations both in the absence and presence of exogenous metabolic activation system. The writing and the flow of the manuscript are acceptable. However, the reviewer has major and minor concerns as follows.
- It is not clear to me what medical device based on substances (MDBS) is tested in the present study. Based on the description in M&M, the MDBS tested is a mixture of several botanical products. Is that the authors purchased all the 8 products and mixed them together according the ratio listed in Table 2? If yes, what is the rationale for the ratio of each product is determined? Or it is one product sold by Aboca?
- Lines 93-94, “For this reason, concentrations equal to or greater than 0.5mg/mL have not been tested in the subsequent experiments in presence of S9 mix.” Metabolic activation may increase the toxicity of a test article, or otherwise has detoxing effect. As 0.25 mg/ml +S9 showed slight toxicity as shown in Fig.1, higher concentrations need to be tested in the presence of S9.
- MN assay, three concentrations were selected, 0.03125, 0.0625 and 0.125mg/mL. Lines 204-205, the authors describe “the OECD guideline n°487 established a cytotoxicity and cytostasis threshold equal to 55±5% and recommends proceeding with the evaluation of genotoxicity”, which is true. In addition, RPD is a recommended means for evaluating genotoxicity by the OECD guideline 487. As shown in Table 1, the RPDs in the 3h+/-S9 treatments at 0.25 mg/ml are greater than 50%, indicating a cytotoxicity of <50%. Accordingly, this concentration needs to be included for conducting the MN assay.
- Table 1, there is no “bar” in the table. In addition, the variation seems big at high concentrations as the data were shown as SEM instead of SD.
- Figure 2, why the x-axis of S9 was ordered differently from the other two figures?
Round 2
Reviewer 2 Report
My concerns have been addressed.